# Identification and Characterization of *Aspergillus nidulans* Mutants Impaired in Asexual Development under Phosphate Stress

**DOI:** 10.3390/cells8121520

**Published:** 2019-11-26

**Authors:** Ainara Otamendi, Eduardo A. Espeso, Oier Etxebeste

**Affiliations:** 1Laboratory of Biology, Department of Applied Chemistry, Faculty of Chemistry, University of The Basque Country, 20018 San Sebastian, Spain; aotamendi005@gmail.com; 2Department of Cellular and Molecular Biology, Centro de Investigaciones Biológicas (CSIC), Ramiro de Maeztu 9, 28040 Madrid, Spain; eespeso@cib.csic.es

**Keywords:** filamentous fungi, mutagenesis, asexual development, conidiation, signal transduction, transcriptional regulation, protein *O*-mannosylation

## Abstract

The transcription factor BrlA plays a central role in the production of asexual spores (conidia) in the fungus *Aspergillus nidulans*. BrlA levels are controlled by signal transducers known collectively as UDAs. Furthermore, it governs the expression of CDP regulators, which control most of the morphological transitions leading to the production of conidia. In response to the emergence of fungal cells in the air, the main stimulus triggering conidiation, UDA mutants such as the *flbB* deletant fail to induce *brlA* expression. Nevertheless, Δ*flbB* colonies conidiate profusely when they are cultured on a medium containing high H_2_PO_4_^−^ concentrations, suggesting that the need for FlbB activity is bypassed. We used this phenotypic trait and an UV-mutagenesis procedure to isolate Δ*flbB* mutants unable to conidiate under these stress conditions. Transformation of mutant FLIP166 with a wild-type genomic library led to the identification of the putative transcription factor SocA as a multicopy suppressor of the FLIP (Fluffy, aconidial, In Phosphate) phenotype. Deregulation of *socA* altered both growth and developmental patterns. Sequencing of the FLIP166 genome enabled the identification and characterization of PmtC^P282L^ as the recessive mutant form responsible for the FLIP phenotype. Overall, results validate this strategy for identifying genes/mutations related to the control of conidiation.

## 1. Introduction

Growth and reproduction are two of the main stages in the life cycle of species, including microbes. However, considering that the resources within a specific niche are finite, species have developed a variety of mechanisms to enable them to disperse to new substrates and to begin a new life cycle. In the case of fungi, the main mechanism of dispersion is the massive production of asexual spores, maximizing the chances that at least one of them will be deposited on a substrate with the right environment for growth.

The kingdom Fungi is composed of 2–4 millions of species [1]. As a consequence, there is a great diversity in the types of asexual developmental structures and spore morphologies generated [2]. It has been suggested that this diversity is caused, at least partially, by different organizations of the genetic and molecular pathways controlling these processes [3,4,5]. Due to an overwhelming number of fungal species, the study of the genetic and molecular mechanisms controlling fungal asexual development has advanced on the basis of a few reference species (see, for example, References [6,7,8]). The filamentous fungus *Aspergillus nidulans* is one such reference species. A majority of the regulators of asexual development that are known were functionally characterized for the first time in this ascomycete.

In general, all asexual spores produced by localized budding and subsequent constriction from an external sporogenous cell are known as conidospores or conidia [2]. This is the case of *A. nidulans*. The external sporogenous cells are known as phialides, but the generation of these secondary sterigmata and the long chains of conidia synthesized by each of them are the final steps of a series of morphological transitions that lead to the generation of a semi-complex developmental structure known as the conidiophore [9,10]. Each conidiophore holds thousands of conidia, and each colony can generate millions of conidiophores under the right culture conditions [11].

There are multiple stimuli inducing the production of conidia in *A. nidulans*. The most important is the emergence of polarly growing cells, the vegetative hyphae, in the aerial environment [10]. According to the accepted model for the genetic control of conidiation in this ascomycete, there are two main pathways controlling the process [6,12,13]. The first, known as the upstream developmental activation (UDA) pathway, is composed of signal transducers that collectively decide whether conidiation is triggered. The second pathway, or central developmental pathway (CDP), controls most stages of conidiophore development. The BrlA (*bristle*) transcription factor links both pathways, since its levels are controlled by UDAs and governs the expression of the CDP regulators.

FlbB, an UDA transcription factor, moves from the growing region of vegetative cells to the nuclei to control in coordination with other developmental regulators the expression of *brlA* [3,14,15,16]. The null mutant of *flbB* fails to induce *brlA* expression when fungal cells emerge into the air [17]. Nevertheless, the development of conidiophores may be triggered by additional stimuli such as light, nutrient starvation, presence of high salt concentrations, alkaline pH, or accumulation of specific metabolites (reviewed in Reference [13]; see also References [18,19,20,21]). Δ*flbB* colonies conidiate profusely when they are cultured on a medium with a high concentration of H_2_PO_4_^−^, suggesting that the need for FlbB activity is bypassed under these conditions.

Supplementation of media with high concentrations of sodium dihydrogen phosphate has been routinely used in our laboratories to promote conidiation in *A. nidulans*. Therefore, the existence of a specific genetic pathway that controls conidiation in response to an excess of H_2_PO_4_^−^ could be possible. To investigate this possibility, we isolated after UV-mutagenesis of the Δ*flbB* strain 80 mutants showing a *fluffy* aconidial phenotype on a growth medium supplemented with phosphate (hereafter referred to as FLIP phenotype, or Fluffy in Phosphate). In this work, we have focused on FLIP166. Sequencing of the FLIP166 genome led to the identification of PmtC^P282L^ as the mutant form responsible for the FLIP phenotype. Also, transformation of FLIP166 protoplasts with a genomic library based on the pRG3-AMA-NotI self-replicating plasmid [22] identified SocA as a multicopy suppressor of the FLIP166 phenotype. Here, we present the functional characterization of the putative transcription factor SocA and the characterization of the PmtC^P282L^ mutation. The developed mutagenesis procedure will lead to the future identification of additional genes related to asexual development and to the updating of the molecular models of this process.

## 2. Materials and Methods

### 2.1. Oligonucleotides, Strains, and Culture Conditions

Strains of *A. nidulans* used in this study are listed in Appendix A, while oligonucleotides used in the generation of transformation constructs or sequencing experiments are listed in Appendix A. The strains were cultivated in either liquid or solid *Aspergillus* minimal (AMM) or complete (ACM) media, adequately supplemented for their respective auxotrophies [23,24]. Glucose (2%) and ammonium tartrate (5 mM) were used by default as sources of carbon and nitrogen, respectively. Nutrient depletion experiments were conducted by diluting the amount of carbon or nitrogen to one-fifth of the original concentration. To evaluate the phenotype of strains under stress conditions, NaH_2_PO_4_ (0.5–1.25 M, initially; 0.65 M thereafter), sucrose (1.0 M), KCl (0.6 M) plus MES (2-(N-morpholino)ethanesulfonic acid, 0.05 M), MgCl_2_ (0.18 M), or H_2_O_2_ (6 mM) were added [17,21]. To analyze the effect of low pH on the phenotype of strains, HCl was used to acidify AMM to 4.23, which is the same pH value of AMM as when 0.65 M NaH_2_PO_4_ is added.

A medium containing 25 g/L corn steep liquor (Sigma-Aldrich, St. Louis, MO, USA) and sucrose (0.09 M) as the carbon source was used as the fermentation medium (AFM) to culture samples for protein extraction [25]. Transformed protoplasts were cultured on selective (lacking uridine and uracil) regeneration medium (RMM: AMM supplemented with 1 M sucrose). Mycelia for DNA extraction and Southern-blot analysis were cultured in liquid AMM. A phenol–chloroform–isoamyl alcohol (25:24:1) mix was used for DNA separation, and the procedures described by us previously were followed [26]. For fluorescence microscopy analyses, conidiospores were incubated for approximately 18 h at room temperature in Ibidi µ-Dishes containing 300 µL of supplemented watch minimal medium (WMM) [27].

Conidia production was quantified as the average of 3–6 replicates per strain. Conidia produced by 72-hour-old colonies were collected in Tween 20 (0.02%). A Thoma cell counter was used to determine the total amount of conidia, which was divided by the area of the colony. The two-tailed Student’s t test for unpaired samples (GraphPad Prism, version 8.0.1, San Diego, CA, USA) was used to determine statistically significant differences in conidia production between the reference strain and mutants. The corresponding column bar graph was drawn using GraphPad Prism.

To characterize, compared to the reference strain, the germination defects of a strain expressing SocA driven by the constitutive *gpdA^mini^* promoter [28], *gpdA^p^*, 10^6^ To characterize, compared to the reference strain, the germination defects of a strain expressing SocA driven by the constitutive *gpdA^mini^* promoter [28], *gpdA^p^*, 10^6^ conidia ml^−1^ were inoculated in Erlenmeyer flasks filled with liquid AMM. Conidia of each strain were incubated at 37 °C and 200 rpm. Samples were analyzed after 6, 8, 10, and 12 h of culture by counting the number of germinated conidia with one or two germ-tubes (from a minimum number of 250 conidia of each strain and time point). Two replicates were analyzed per strain. GraphPad Prism was used to draw the corresponding graph.

.ml^−1^ were inoculated in Erlenmeyer flasks filled with liquid AMM. Conidia of each strain were incubated at 37 °C and 200 rpm. Samples were analyzed after 6, 8, 10, and 12 h of culture by counting the number of germinated conidia with one or two germ-tubes (from a minimum number of 250 conidia of each strain and time point). Two replicates were analyzed per strain. GraphPad Prism was used to draw the corresponding graph.

DNA constructs for transformation were generated by the fusion-PCR technique [29]. For 3´-end gene tagging (wild-type or mutant versions), (1) approximately 1.5 Kb of the 3´-end of the coding region (oligonucleotides GSP1 and GSP2; Appendix A), (2) a second fragment containing the tag (*gfp* or *ha_3x_*) plus the selection marker (*pyrG^Afum^* of *A. fumigatus*; oligonucleotides GFP1 and GFP2), and (3) 1.5 Kb of the 3´-UTR region of the gene (oligonucleotides GSP3 and GSP4) were amplified and subsequently fused. For gene deletion, the same procedure was followed with (1) approximately 1.5 Kb of the promoter region of the gene (oligonucleotides PP1 and PP2), (2) the selection marker (SMP1 and GFP2), and (3) 1.5 Kb of the 3´-UTR region of the gene (GSP3 and GSP4). To generate strains expressing *gpdA^p^*-driven SocA::GFP or SocA::HA_3x_ chimeras, five fragments were amplified and fused essentially as described by us previously [14]: (1) 1.5 Kb of the promoter (oligonucleotides PP1 and PP2´-ATG), (2) *gpdA^p^* (gpdAUp and gpdADw), (3) the coding region of *socA* (geneSP and GSP2), (4) *gfp* or *ha_3x_* plus *pyrG^Afum^* (GFP1 and GFP2), and (5) 1.5 Kb of the 3´-UTR region (GSP3 and GSP4). Generation and transformation of protoplasts of recipient strains was done as described by Tilburn and colleagues as well as Szewczyk and colleagues [30,31].

Meiotic crosses were carried out essentially as described by Todd et al. [32]. Diploids were generated by inoculating on selective AMM conidia of the corresponding heterokaryons [33] or by culturing on selective RMM a mix of protoplasts of the parental strains. Haploidization of diploids was induced by adding benomyl (1–3 µg/mL) to the culture medium [34]

### 2.2. Mutagenesis Procedure

To mutagenize Δ*flbB* conidia, these were collected from colonies previously cultured for 96 h at 37 °C on AMM supplemented with 0.65 M NaH_2_PO_4_. Approximately 1250 conidia were inoculated on Petri plates of 14-cm diameter filled with the same culture medium. Conidia (all plates but one, which was used as control) were subjected to 254-nm UV radiation using a Mineralight Lamp (Model UVGL-58 lamp; intensity: 1350 µW/cm^2^). Exposure times were selected to cause a survival rate of 5–10%, usually 80–100 s [17,26]. Plates were cultured at 37 °C for 72 h, and mutants unable to conidiate were transferred to 5.5-cm plates filled with AMM plus 0.65 M NaH_2_PO_4_ before phenotypic characterization.

### 2.3. Cloning of socA

*An8501*/*socA* was cloned after transformation of FLIP166 protoplasts with a sample of a wild-type gene library cloned in the self-replicating plasmid p*RG3-AMA-NotI* [22]. Revertants were identified as conidiating colonies, which were grown to isolate genomic DNA plus the corresponding plasmids by transformation in *E. coli*. Purified plasmids were tested for complementation of the FLIP166 aconidial phenotype, and oligonucleotides pRG3Up and pRG3Dw were used to sequence flanking regions of inserts. These sequences were used to delimit the full genomic sequence in the insert by comparison with the *A. nidulans* genome database.

### 2.4. Bioinformatics

The Integrative Genomics Viewer (IGV; [35]) was used to visualize DNA- and RNA-seq reads and to correct the predicted coding sequence of *socA*. Prediction of the presence of functional domains in protein sequences was done using InterPro [36]. Query sequences were *BLASTed* using NCBI (https://blast.ncbi.nlm.nih.gov/Blast.cgi) or EMBL-EBI (https://www.ebi.ac.uk/Tools/sss/ncbiblast/) websites. Clustal omega was used for sequence alignment (https://www.ebi.ac.uk/Tools/msa/clustalo/) while the open software Genedoc (version 2.7.000) was selected for their visualization. Phylogenetic trees were constructed with MegaX (version 7.0) [37] and the neighbor-joining method, with a minimum bootstrap of 1000 replicates. The trees were edited using iTOL (Interactive Tree of Life, version 5.2) [38]. To build a structure homology model for PmtC, Swiss Model [39] and Phyre2 (version 2.0) [40] servers were used. Finally, DynaMut was used to assess the impact of specific mutations on protein conformation [41].

### 2.5. Fluorescence Microscopy

Subcellular localization of SocA and PmtC was analyzed using a Leica DMI-6000b inverted microscope following our standard procedures [14,15]. The microscope is equipped with a 63 × Plan Apo 1.4 N.A. oil immersion lens from Leica, and a GFP filter (excitation at 470 nm and emission at 525 nm). ImageJ was used to process fluorescence microscopy and phase contrast images (https://imagej.nih.gov/ij/) (US. National Institutes of Health, Bethesda, MA, USA).

### 2.6. Protein Extraction and Immunodetection

Mycelial samples of strains expressing the chimera SocA::GFP were grown by default in AFM for 15 h at 37 °C and 200 rpm. To analyze the effect of phosphate, 10^6^ conidia/mL were inoculated in Erlenmeyer flasks filled with 25 mL of liquid AFM, cultured for 15 h at 37 °C and 200 rpm, filtered, and transferred to AMM supplemented with 0.65 M H_2_PO_4_^−^. Samples for protein extraction were collected after 0, 15, 30, 45, 60, 120, 180, and 240 min of culture and lyophilized.

For total protein extraction, we followed the protocol described by Hervás-Aguilar and Peñalva [42]. Cell lysis was carried out in 1 mL of alkaline lysis buffer (0.2 M NaOH and 0.2% β-mercaptoethanol), and proteins were precipitated by adding 10% trichloroacetid acid (TCA). Samples were centrifuged, and the supernatant was discarded. Pellets were resuspended in 100 μL Tris-Base (1 M) and 200 μL of SDS-PAGE loading buffer (62.5 mM Tris–HCl pH = 6.8, 2% SDS (*p*/*v*), 5% β-mercaptoethanol (*v*/*v*), 6 M urea, and 0.05% bromophenol blue (*p*/*v*)). Aliquots of 10 µl were then loaded and separated in 10% polyacrylamide gels [43] before transference to nitrocellulose filters. For the immunodetection of SocA::GFP, an α-GFP (mouse) monoclonal antibody cocktail (1:5000 Roche) was used. Peroxidase-conjugated α-mouse (1:4000, Jackson Immuno Research Laboratories) was used as secondary antibody. The Amersham Biosciences ECL kit was used to induce peroxidase activity, and chemiluminescence was detected using a Chemidoc + XRS system (Bio-Rad, Hercules, CA, USA).

## 3. Results

### 3.1. Isolation of FLIP (Fluffy in Phosphate) Mutants

Null and loss-of-function *flbB*^−^ colonies display a characteristic *fluffy* aconidial phenotype on AMM (Figure 1A; left). This aconidial phenotype can be reverted, nonetheless, by culturing on AMM supplemented with an elevated concentration of sodium phosphate, specifically NaH_2_PO_4_. To identify additional genes related to the control of conidiation in *A. nidulans* and/or genes potentially involved in a pathway inducing asexual development in response to high phosphate concentrations, we mutagenized Δ*flbB* conidia and isolated mutants unable to conidiate on AMM containing 0.65 M NaH_2_PO_4_ (Figure 1B). Short-wave (254 nm) UV mutagenesis was carried out as indicated in the Materials and Methods section using exposure times allowing only 5–10% of Δ*flbB* conidia to survive (Figure 1B). Among 11,000 survivors, eighty mutants showing a FLIP phenotype were selected. Their aconidial phenotypes were compared to that of the null *flbB* parental strain on AMM and AMM plus 0.65 M NaH_2_PO_4_. Figure 1C shows a selection of FLIP mutants which displayed different conidiation and/or colonial growth deficiencies.

### 3.2. Phenotypic Characterization of FLIP Mutants

First, we considered the possibility of FLIP strains carrying mutations in previously characterized regulators of asexual development. With this aim, we compared growth on solid AMM and AMM plus 0.65 M NaH_2_PO_4_ of a set of FLIP mutants representative of the different phenotypic groups to strains carrying null alleles of genes coding for UDA regulators and also the Δ*brlA* strain (Appendix A). As a result, potential mutations in UDA genes *flbC*, *flbE*, *tmpA*, *tmpB*, and *fluG* were discarded for the FLIP mutants. The Δ*flbC* strain produced on AMM abundant Hülle cells after 72 h of culture. The Δ*tmpA* and Δ*tmpB* strains conidiated profusely in both media. Finally, the Δ*flbE* and Δ*fluG* strains conidiated on AMM supplemented with 0.65 M NaH_2_PO_4_. Although, as expected, the Δ*brlA* strain showed a severe aconidial phenotype under phosphate stress, it was also discarded after a detailed microscopic analysis (not shown). Similarities were observed when comparing FLIP mutants with a Δ*flbD* strain and the double null mutant Δ*flbB*;Δ*flbD* (see mutants FLIP121 and FLIP196 in Appendix A). The transformation of protoplasts of FLIP121 and an additional two FLIP mutants with similar phenotypes (FLIP33 and FLIP104) with an *flbD*::*gfp*::*pyrG^Afum^* construct did not restore a Δ*flbB* phenotype on AMM medium with 0.65 M NaH_2_PO_4_ (Appendix A). Thus, we ruled out the possibility of these three mutants being double loss-of-function Δ*flbB*;*flbD*^−^ mutants.

Second, we classified the 80 FLIP mutants into groups according to their main phenotypic traits. Hence, in addition to AMM and AMM plus 0.65 M NaH_2_PO_4_, the effect of carbon or nitrogen limitation (one-fifth of the quantity added to standard AMM) and the addition of sucrose (1.0 M), H_2_O_2_ (6 mM), KCl (0.6 M; buffered at pH 6.5 with 0.05 M MES), or MgCl_2_ (0.18 M) were analyzed. Again, the parental Δ*flbB* mutant and the Δ*flbC*, Δ*flbD*, and Δ*brlA* strains were used as references. Six phenotypic groups were differentiated among FLIP mutants (Figure 2). The first one (20 mutants out of 80; 25%) included those showing a Δ*flbD*-like phenotype on AMM with 0.65 M NaH_2_PO_4_, which produced conidia in the center of the colony (FLIP33, FLIP99, and FLIP121 in Figure 2; note the phenotypic differences among these three mutants and compared to a Δ*flbD* strain on AMM with 1.0 M sucrose; see also Appendix A). The second group included FLIP mutants with a clearly aconidial phenotype on AMM with 0.65 M NaH_2_PO_4_ (19 out of 80; 23.8%; i.e., FLIP51, FLIP144, FLIP166, FLIP176, and FLIP196 in Figure 2). The third group comprised FLIP mutants showing a reduced colony diameter (18 out of 80; 22.5%; i.e., FLIP16 and FLIP94). Mutants in group 4 displayed a reduced-radial growth phenotype that was reverted under specific culture conditions (5 out of 80; 6.3%; see, for example, FLIP149 and mainly FLIP204 on medium with 0.65 M NaH_2_PO_4_). Those FLIP mutants with a wet, purportedly autolytic, phenotype in the center of the colony were included into the fifth group (8 mutants out of 80; 10%; see, for example, FLIP199 in AMM with KCl/MES). Finally, the sixth group included the rest of FLIP mutants which could not be included in any of the five groups described above (10 out of 80; 12.5%; not shown).

The addition of high concentrations of H_2_PO_4_^−^ caused a decrease in the pH of the extracellular medium. To evaluate an effect of medium acidification on the phenotypes scored, FLIP mutants were cultured on AMM adjusted to pH 4.23 with HCl (Appendix A). For most FLIP mutants and the parental Δ*flbB* strain, results suggested that the phenotype observed in AMM with 0.65M NaH_2_PO_4_ was not caused exclusively by the acidification of the culture medium.

### 3.3. An8501/socA as a Multicopy Suppressor of the FLIP166 Phenotype: Functional Characterization

To investigate dominance of FLIP mutations, diploid strains were generated between specific FLIP mutants and strain RMSO11 (see Appendix A; see also Appendix A). Benomyl haploidization confirmed the diploid nature of the strains generated, while their wild-type phenotype confirmed the recessive nature of the mutations causing the FLIP phenotype of mutants 16 and 166.

Transformation of FLIP166 protoplasts with a wild-type pRG3-AMA-NotI library led to the isolation of transformants that were able to conidiate on selective culture medium supplemented with 0.65 M NaH_2_PO_4_ (Appendix A). Plasmids causing the phenotypic reversion were recovered, and their inserts were sequenced, leading to the identification of *An8501*. As expected, due to the use of a wild-type library, plasmids bearing a copy of *flbB* were also recovered (see below). Sequencing of *An8501* led to the identification of a single nucleotide polymorphism in the fourth predicted intron (Figure 3A) [44]. Thus, we concluded that *An8501* is a multicopy suppressor of the FLIP166 phenotype and consequently named it *socA* (suppressor of the FLIP conidiation phenotype). An incorrect annotation of the coding region of *socA* was also noticed (Figure 3A), since RNAseq reads clearly showed that the first intron is shorter in 5′. This made us modify the prediction for the first exon and intron, adding in the former codons for amino acids that are important for the prediction of the functional domains of SocA.

Detection of a Zn2-Cys6 binuclear zinc cluster (IPR001138) in the N-terminus and a Fungal transcriptional regulatory domain (IPR021858) in the C-terminal half of SocA (Figure 3A) strongly supports a regulatory role for SocA. BLAST analyses using as query the corrected sequence of SocA identified putative orthologs in Eurotiales and Onygenales orders within the class of Eurotiomycetes (Figure 3B). Interestingly, the proteomes of multiple Aspergillaceae species (with exceptions such as, for example, *A. nidulans* itself) included two paralogs of SocA, which were mapped to different clades of our tree. Overall, phylogenetic analyses suggest that SocA emerged within Eurotiomycetes and that its evolutionary history included at least one duplication event.

A functional characterization of SocA was done to investigate its role as a regulator of conidiation. Strains carrying the null *socA* allele and strains expressing SocA::GFP driven either by the native or the constitutive *gpdA^p^* promoter were generated, each of them in both wild-type and Δ*flbB* backgrounds (Figure 4A). The pattern of conidiation was unaltered in Δ*socA* and SocA::GFP strains in the wild-type *flbB* background. Significant differences were observed, however, in the Δ*flbB* background on AMM supplemented with 0.65 M of NaH_2_PO_4_. After 48 h of culture, conidia production by the double-null Δ*flbB*;Δ*socA* strain decreased compared to the parental Δ*flbB* strain, from 5.54 × 10^6^ ± 0.22 × 10^6^ to 0.53 × 10^6^ ± 0.19 × 10^6^ conidia/cm^2^ (*p* < 0.0001, significant change; n = 6 for each strain; Figure 4B). The Δ*flbB*;*socA*::*gfp* strain showed an intermediate phenotype compared to the double null and the parental Δ*flbB* mutants, although conidia quantification showed a statistically nonsignificant change in asexual spore production compared to the null *flbB* strain (5.63 × 10^6^ ± 0.47 × 10^6^ conidia/cm^2^; *p* = 0.84; n = 3).

Constitutive and higher expression of SocA::GFP driven by the *gpdA^p^* promoter was confirmed through immunodetection in Figure 4C (the Coomassie-stained gel is shown as a control of equal loading of the protein samples). This caused a strong inhibition of radial growth in all conditions and backgrounds tested (Figure 4A), suggesting that misscheduled and elevated levels of SocA probably have detrimental effects on colony growth. Additional immunodetection experiments using protein extracts of mycelial samples grown in liquid medium supplemented with 0.65 M of NaH_2_PO_4_ suggested that *socA* expression is not induced under these stress conditions (Figure 4D).

As expected for a putative transcription factor, fluorescence microscopy analyses showed a nuclear localization of the *gpdA^p^*-driven SocA::GFP chimera (Figure 4E). These analyses also revealed an altered germination pattern in this strain (arrowheads in Figure 4E), suggesting that the reduced radial growth phenotype caused by constitutive expression of *socA* was probably a consequence of a delayed germination and an altered polarity establishment. The graph in Figure 4F confirms that germination in this strain is delayed (see the dotted line). For example, while the *gpdA^p^*-driven SocA::HA_3x_ strain needs 12 h to achieve a 73% germinated conidia, the reference strain would need less than 9 h (n = 2 measurements per strain).

### 3.4. The Mutant Form Pro282Leu of the Protein O-Mannosyltransferase PmtC Is Responsible for the FLIP166 Phenotype

Transformation of FLIP166 protoplasts with a p*RG3*::*flbB* plasmid isolated from the wild-type library showed that the unknown mutation in the FLIP166 mutant caused an aconidial phenotype on AMM and that this aconidial phenotype was reverted on AMM supplemented with 0.65 M NaH_2_PO_4_ (Figure 5A). This observation suggested that the gene mutated in FLIP166 is not a member of a hypothetic pathway signaling conidiation exclusively in response to phosphate stress and that the FLIP166 phenotype is an additive effect of *flbB* deletion and the unknown mutation.

As a first approach to its identification, we sequenced the genome of the FLIP166 mutant (Stabvida; Caparica, Portugal) and focused the analysis on exonic mutations compared to 1) the FGSCA4 genome, used as the reference in the IGV software; 2) RNAseq reads of a Δ*flbB* strain and its isogenic wild-type strain [44,45]; and 3) RNAseq reads of a Δ*sltA* strain and its reference wild-type strain (E.A. Espeso, unpublished). Up to ten candidate exonic mutations were identified in FLIP166 genomic sequence compared to the genomes of reference strains and transcriptomic data (Appendix A). Seven of these candidate mutations were sequenced in FLIP166 and the parental Δ*flbB* strain. Only the hypothetic mutation in An6835 was discarded, while the rest were confirmed as bona fide single nucleotide variations (SNVs) in the FLIP mutant.

To delimit the list of candidates, meiotic crosses of FLIP166 were carried out and diploids were generated with the master strains FGSCA68 and FGSCA283 (Appendix A). The wild-type phenotype of the diploid strains confirmed again that the mutation responsible for the FLIP166 phenotype is recessive, while an analysis of the meiotic progeny, on the one hand, and the haploids isolated using benomyl, on the other hand, focused our search on chromosome VII (not shown). Since we obtained in meiotic analyses a high percentage (>85%) of acriflavine sensitive colonies, we also took chromosome II into consideration. Thus, the search was limited to three candidate mutations in the following genes (Appendix A): (1) an ATT (Ile) to AAT (Asn) point substitution in codon 94 of *An12172* (Chr II), coding for a putative LCCL domain-containing transmembrane protein; (2) a CTG (Leu) to CGG (Arg) point mutation in codon 99 of *An12335/acdA* (Chr VII), coding for a peroxisomal acyl-CoA dehydrogenase [46]; and (3) a CCG (Pro) to CTG (Leu) point mutation in codon 282 of *An1459/pmtC* (Chr VII), coding for a protein O-mannosyltransferase involved in hyphal growth and conidia formation [47,48,49].

DNA constructs consisting of *gfp*-tagged wild-type versions of each of the three genes were used in transformations to determine which SNVs in FLIP166 were responsible for its phenotype. Only the targeted insertion of the *pmtC*::*gfp*::*pyrG^Afum^* construct at its native *locus* caused the reversion of the FLIP166 phenotype on AMM supplemented with 0.65 M NaH_2_PO_4_ (Figure 5B). Reversely, replacement of wild-type *pmtC locus* in the parental Δ*flbB* strain with the *pmtC^Pro282Leu^*::*gfp*::*pyrG^Afum^* construct caused an aconidial phenotype on AMM with 0.65 M NaH_2_PO_4_, similar to that observed in FLIP166. Such an aconidial phenotype was not observed when this parental strain was transformed with *An12172^Ile94Asn^*::*gfp*::*pyrG^Afum^* or *acdA^Leu99Arg^*::*gfp*::*pyrG^Afum^* constructs (Figure 5B). Overall, our results show that the *pmtC^Pro282Leu^* mutation is responsible for the aconidial FLIP166 phenotype on AMM supplemented with 0.65 M NaH_2_PO_4_. Nevertheless, we observed that the radial growth of FLIP166 was bigger than that of the Δ*flbB* strain transformed with the *pmtC^Pro282Leu^*::*gfp*::*pyrG^Afum^* construct, suggesting that the presence of indeterminate mutations in FLIP166 may be also affecting radial growth.

Furthermore, in this set of transformations, we isolated two Δ*flbB* strains that, in the absence of the mutation corresponding to the Pro282Leu substitution, presented other mutations in *pmtC*. One of them caused the Arg762Gly point substitution while the second one introduced a premature stop at position 767 (Tyr767Stop). Both mutations generated a FLIP aconidial phenotype on medium supplemented with 0.65 M NaH_2_PO_4_ (Figure 5B) but not an inhibition of radial extension at the level of the Pro282Leu mutation. These results strongly suggest that the C-terminal end of PmtC is necessary for the control of conidiation but may be less important in the control of growth.

### 3.5. Substitution of Proline in Position 282 of PmtC Causes an Aconidial Phenotype and an Inhibition of Growth but Not a Delocalization of the Protein

PmtC is a protein O-mannosyltransferase of the subfamily 4. Protein O-mannosyltransferases initiate glycosylation of proteins in the lumen of the endoplasmic reticulum (ER) by transferring a mannosyl residue from dolichyl-P-mannose to the hydroxyl group of Ser or Thr amino acids of target proteins [48,50,51]. Then, additional carbohydrate residues are added to this mannose in the Golgi apparatus. There are 3 protein O-mannosyltransferases in *A. nidulans*, all of them conserved in fungi and also outside the fungal kingdom [51]. Our phylogenetic tree, built with 2169 fungal orthologs of the three Pmt proteins of *A. nidulans*, represents most of the classes of the fungal tree of life (Figure 6) and suggests that all fungal classes have conserved the three Pmt orthologs, probably due to their central roles in posttranslational modification of proteins.

Our RNAseq results [44,45] confirmed the previous annotation of the coding region of *pmtC* in the AspGD and FungiDB databases. *pmtC* gene structure is shown in Appendix A. InterPro searches predicted the presence of three characteristic domains in PmtC (Figure 7A): First, a glycosyltransferase 39-like domain (IPR003342) between residues 49 and 293, which would be responsible for the mannosyltranferase activity; second, an MIR domain (IPR016093) between residues 324 and 514, with three repetitions of the MIR motif which is predicted to form a closed β-barrel structure that may have a ligand transferase function; and finally, a protein O-mannosyltransferase C-terminal domain (IPR032421) between residues 536 and 765, which would add four transmembrane domains to the protein.

Protein structure homology-modeling of PmtC in SwissModel and Phyre2 servers [39,40] predicted the presence of eight transmembrane (TM) helices along the N-terminal glycosyltransferase domain plus the remaining four of the C-terminal end of the protein, totaling 12 membrane-pass helices (Figure 7A). The point mutation Pro282Leu was located within the eighth TM helix (see also the location of the Arg762Gly substitution in the C-terminus of the protein). Three-dimensional modeling of PmtC was based on Protein Data Bank (PDB) entry 6P25, and 96% of residues were modeled at a confidence higher than 90%. Results indicate that all TM helices group together in an α-structure that would be inserted in the ER membrane while the β-barrel corresponding to the MIR domain would be oriented to the lumen of the ER (Figure 7A,B; see also Reference [51]). DynaMut [41] predicted that Pro282Leu substitution increases rigidity of the eighth TM helix of PmtC (stabilizing) as well as hydrophobic interactions of the mutant Leu residue with Phe248 (TM7) (Figure 7C, top row). In the case of the Arg762Gly substitution, the insertion of a Gly residue is predicted to destabilize the protein in the C-terminus of the last TM helix (gain of flexibility) (Figure 7C, bottom row).

Protoplasts of a wild-type strain were transformed with wild-type or mutant (Pro282Leu) *pmtC*::*gfp*::*pyrG^Afum^* constructs. Expression of the mutant PmtC^Pro282Leu^::GFP form, but not that of the wild-type version, caused a strong inhibition of radial growth and conidiation (Figure 8A), as shown before for the null mutant [47,49]. This phenotype, in contrast to that shown by FLIP166 and the Δ*flbB* strain expressing PmtC^Pro282Leu^::GFP (see before), suggests that the deletion of *flbB* suppresses partially the growth defects caused by the substitution by a leucine of proline in position 282 of PmtC (see the Discussion section).

Despite its low fluorescence levels, localization of wild-type PmtC was determined for the first time. PmtC localized, in both wild-type and Δ*flbB* backgrounds, to structures resembling the endoplasmic reticulum (Figure 8B). As was described previously for ER proteins such as Sec63 and Sec12 [52,53], nuclear envelopes (NE in Figure 8B) as well as a faint network of interconnected tubules and patches (P) could be discerned in correlation with the predicted function of PmtC as a protein O-mannosyltransferase localized in the ER. Indeed, the induction of the unfolded protein response (UPR) and inhibition of growth triggered by the addition of 8 mM dithiothreitol (DTT) was accompanied by a relocalization of PmtC to bright and bigger spots (Figure 8C), as was described previously by Markina-Iñarrairaegui and colleagues for Sec63 [52]. Removal of DTT from the culture medium restored growth and the native localization of PmtC (Figure 8C, bottom).

Hyphae of the mutant Pro282Leu strain showed a hyperbranching phenotype that explained, at least partially, the phenotype described in Figure 8A (both pictures in Figure 8D correspond to the same hyphal networks analyzed in more detail in Figure 8B, left, and Figure 8E, left). Interestingly, we did not observe delocalization of PmtC due to Pro282Leu (wild-type and Δ*flbB* backgrounds) or Arg762Gly (Δ*flbB* background) substitutions (Figure 8E). These results suggest that residues Pro282 within TM helix 8 and Arg762 within the protein O-mannosyltransferase C-terminal domain are necessary for PmtC activity but may not be essential for its cellular localization.

## 4. Discussion

### 4.1. A Novel Mutagenesis Strategy as a Tool for the Identification of Hitherto Unknown Genes and Mutations Related to the Control of Conidiation

In their seminal works on *A. nidulans* conidiation, Timberlake proposed that more than a thousand different mRNAs accumulate to varying concentrations, specifically during this developmental process, while Martinelli and Clutterbuck reduced drastically the number of uniquely required genes to 45–100 [10,54,55]. In a more recent RNA-sequencing analysis, Garzia and collaborators described that approximately 2200 genes were significantly deregulated when RNA samples collected during vegetative growth and five hours after the induction of conidiation were compared [45]. More than 90% of them (2035 out of 2222) were significantly downregulated, while only a small fraction (187) was upregulated, introducing a consensus view for the abovementioned discrepancy. Although the number of functionally characterized direct or indirect regulators of conidiation is growing, it can be stated that we are still far from seeing the big picture on the mechanisms controlling asexual development and how they are coordinated with other fundamental cellular processes such as polar growth, sexual development, and the response to stress conditions. FlbB is one of these key regulators of conidiation, and here, we have added a collection of Δ*flbB* mutants unable to produce conidia under stress caused by high concentrations of H_2_PO_4_^−^. We have identified and carried out a preliminary functional characterization of SocA and have extended our knowledge on the localization and the role in conidiation of PmtC [47,48,49]. The future characterization of additional FLIP mutants may lead to the identification of new mutations in known regulators and/or the identification of hitherto unknown regulators of conidiation, furthering a deeper understanding and the updating of the current genetic and molecular models for the control of asexual sporulation in this ascomycete. Some of these genes may control *brlA* expression specifically in response to phosphate stress. In this work, we have sequenced *pmtC* in six FLIP mutants from phenotypic groups 2 and 3 (not shown) and all of them bore wild-type versions of the gene, suggesting that the mutations triggering the FLIP phenotype of these strains are not loss-of-function versions of *pmtC*. We are currently identifying these mutations.

### 4.2. SocA, a Putative Transcriptional Regulator Probably Involved in the Control of Morphogenesis and Development under Specific Stress Conditions

*An8501*/*socA* has been identified here as a multicopy suppressor of aconidial FLIP166 and *flbB*^−^ phenotypes. Our RNA-seq results indicate that there is no significant deregulation of its expression five hours after triggering asexual differentiation (not shown). The expression of specific UDA regulators of conidiation is increased at this time point, just before the upregulation of *brlA* [16,17]. This observation and the wild-type phenotype of null *socA* mutant (wild-type *flbB* background) in AMM suggest that this putative transcription factor is not directly involved in the regulation of *brlA* expression in particular and in the control of conidiation in general.

SocA was not included in the list of Zn2Cys6-type transcription factors after the update of the *A. nidulans* genome annotation [56] most probably due to the annotation error noticed here in the first exon and intron. Our evolutionary analysis of the corrected amino acidic sequence of SocA suggests that it emerged with the class of Eurotiomycetes, probably, as a duplication of an additional regulator including both the Zn finger (N-terminal) and the fungal transcriptional regulatory domain (C-terminal). Interestingly, multiple *Aspergilli* and *Penicilli* have a paralog of SocA. It is unknown whether this duplication has led to neofunctionalization (the development of new functions) or subfunctionalization (retention of a subset of the functions of the original paralog) events [57,58] or if any of these paralogs have retained in these species the functional relationship with conidiation described in the present work for *A. nidulans* SocA. Functional characterization of these paralogs is necessary to answer these questions.

SocA levels are not affected by the addition of 0.65 M H_2_PO_4_^−^ to AMM, and conidia production by the null *socA* strain (wild-type background) is not inhibited in these conditions. These results suggest that SocA is not a member of a hypothetic pathway required for the developmental response to phosphate stress. Nevertheless, the phenotype of the Δ*socA* strain is completely different in a Δ*flbB* background (also, that of the *socA*::*gfp* strain, highlighting the importance of the C-terminally located transcriptional regulatory domain in this response), with a clear inhibition of conidia production to the point of resembling a FLIP mutant of group 1. In this context, it has been proposed that, as well as the insertion of an additional regulator within a preexisting transcriptional network can cause its rewiring [58], deletion of a gene coding for a transcription factor can also trigger rewiring events [59]. Finally, constitutive upregulation of *socA* delays growth and modifies the germination pattern, suggesting that its levels must be precisely adjusted to optimize growth. Overall, these observations and the low *socA* expression levels before and after the induction of conidiation both in wild-type and Δ*flbB* backgrounds suggest that this putative transcription factor is required under specific (and unknown) stress conditions and that its misscheduled deregulation alters growth and developmental patterns.

### 4.3. Revisiting the Function of PmtC

Protein O-mannosyltransferases are involved in the mannosylation of target proteins at serine and threonine residues [51]. That is supposed to occur in the ER, while the addition of the rest of the carbohydrates to this mannosyl residue, conforming the final glycosylaton step, is carried out in the Golgi apparatus. This role correlates with the localization of PmtC::GFP described in this work, which is not altered in Pro282Leu or Arg762Gly mutants, suggesting that the loss of function is not due to a delocalization of the protein. The ER–Golgi network is polarized in *A. nidulans*, and it organizes and delivers to the apical membrane (growth of fungal hyphae is polarized and the growth site is known as the apex) plasma membrane and cell-wall components (see, for example, Reference [60]). In this context, the inhibition of growth observed in the strain expressing the mutant PmtC^Pro282Leu^ form was expected and is associated to a role of PmtC in the mannosylation of proteins involved in polar growth. For example, Kriangkripipat and Momany described that PmtC (and PmtA) modifies the putative plasma-membrane stress sensor and transducer WscA/An5660 [48]. Deletion of *wscA* inhibits significantly radial growth of *A. nidulans* colonies [61].

Interestingly, the growth phenotype observed as a consequence of introducing a Pro282Leu substitution in PmtC in the wild-type background is suppressed, at least partially, by deletion of *flbB*. We have previously observed in a Δ*flbB* background a reversion of the growth defects caused by the deletion of the myosin-coding gene *myoB* (our unpublished results). These results suggest that there is a genetic link between the control of conidiation and polar growth. This hypothetic connection should be analyzed in the future.

The inhibition of conidiation observed in PmtC^Pro282Leu^ mutants may be indirect due to the participation of PmtC in the mannosylation of a protein required for both growth and conidiation. It must be remembered that specific stages of conidiophore development, such as stalk or metulae emergence, are based on apical growth and are controlled by regulators of polarity [9,62]. However, PmtC might be involved directly in the mannosylation/glycosylation of a regulator of conidiation that transits the ER–Golgi network and, maybe, reaches the hyphal apex. The only known regulators of conidiation in *A. nidulans* showing an apical localization are FlbB and FlbE [14,15,26]. However, it is unlikely a direct role of PmtC in the posttranslational modification of FlbB. In that case, the FLIP166 mutant would show a conidiating phenotype in AMM supplemented with 0.65 M H_2_PO_4_^−^ as does the Δ*flbB* mutant and not an additive, completely aconidial phenotype. Thus, we predict that a hypothetic target of PmtC controlling conidiation should be a protein other than FlbB.

It has been proposed that Pmt4/PmtC forms not only homodimers in *A. nidulans* but also heterodimers with PmtA and PmtB [63]. Thus, mutations in the corresponding genes *pmtA* and *pmtB* will likely induce or contribute to the FLIP phenotype and may be identified in the study of the library of FLIP mutants available. Overall, the functional characterization of mutant forms of these or other hitherto unidentified proteins will deepen our understanding of how stimuli are transduced and integrated to coordinate morphogenetic, developmental, and viability processes.

## Figures and Tables

**Figure 1 cells-08-01520-f001:**
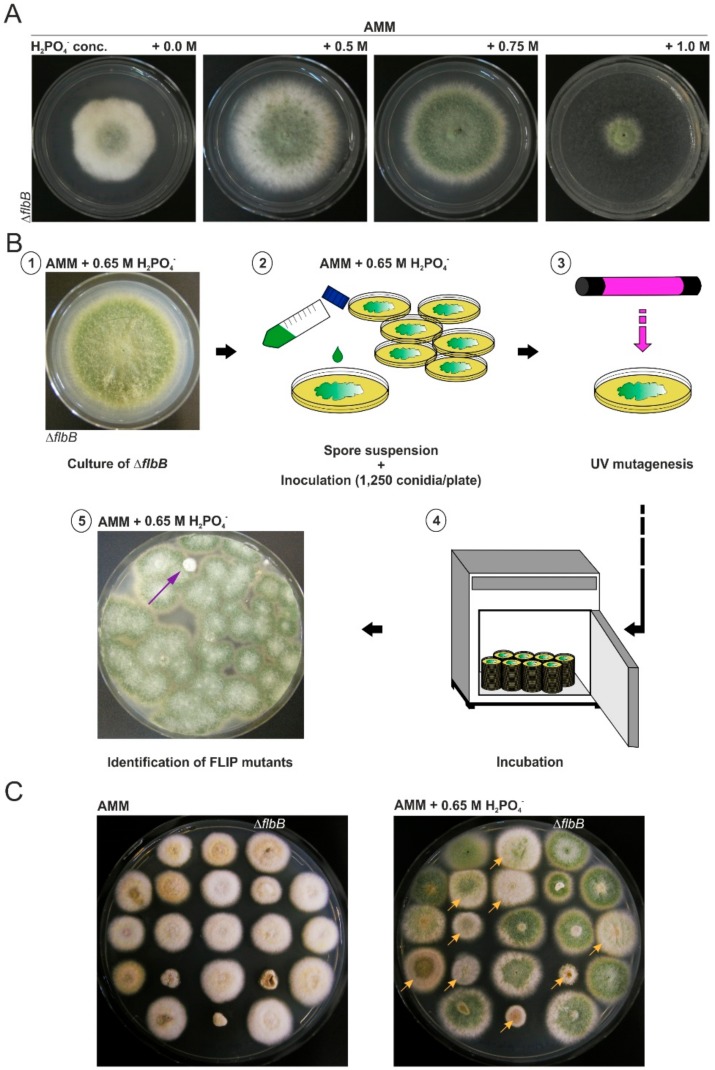
Mutagenesis procedure followed to isolate FLIP (Fluffy in Phosphate) mutants: (**A**) Phenotype of the parental Δ*flbB* strain on *Aspergillus* minimal medium (AMM) supplemented with increasing concentrations of NaH_2_PO_4_. Petri dishes (diameter: 5.5 cm) were cultured at 37 °C for 72 h. A concentration of 0.65 M was established. (**B**) Mutagenesis procedure followed for the isolation of FLIP mutants: Δ*flbB* conidia were collected and inoculated (1250 conidia/plate; diameter: 14 cm) on AMM dishes supplemented with 0.65 M NaH_2_PO_4_. Conidia (all plates but one, which was used as negative control) were mutagenized by using UV radiation of 254 nm and exposure times of 80–100 s. After 72 h of culture at 37 °C, mutants unable to conidiate were transferred to 5.5-cm dishes filled with AMM plus 0.65 M NaH_2_PO_4_ before phenotypic characterization. (**C**) Growth and conidiation phenotypes of a group of FLIP mutants in AMM and AMM plus 0.65 M H_2_PO_4_^−^ compared to the parental null *flbB* strain and after 72 h of incubation at 37 °C (diameter of Petri dishes: 9 cm). The arrows designate FLIP mutants with different phenotypes (see also Figure 2).

**Figure 2 cells-08-01520-f002:**
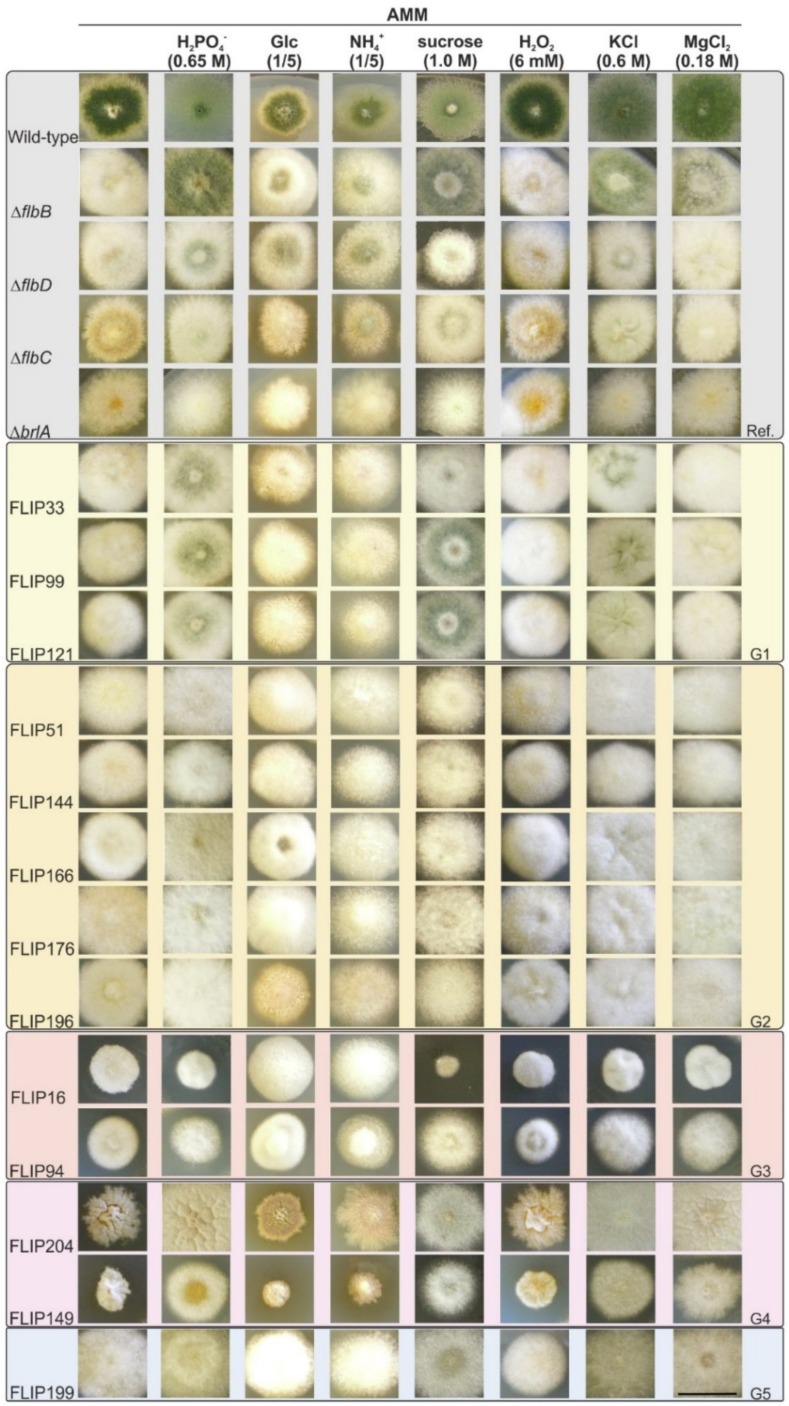
Phenotypic classification of FLIP mutants: Phenotype of specific FLIP mutants after 72 h of culture at 37 °C on AMM, on AMM supplemented with NaH_2_PO_4_ (0.65 M), sucrose (1.0 M), KCl (0.6 M) plus MES (0.05 M), MgCl_2_ (0.18 M), or H_2_O_2_ (6 mM), as well as on AMM in which carbon or nitrogen sources were diluted to one-fifth of the original concentration. The different FLIP phenotypes were compared to reference strains with a wild-type or aconidial (parental Δ*flbB*; Δ*flbD*, Δ*flbC*, and Δ*brlA*) phenotypes. Five (G1–5) out of the six phenotypic groups are shown: G1—FLIP mutants with a Δ*flbD*-like phenotype on AMM with 0.65 M NaH_2_PO_4_ (FLIP33, FLIP99, and FLIP121 are shown; see also Appendix A); G2—FLIP mutants with a clearly aconidial phenotype on AMM with 0.65 M NaH_2_PO_4_ (i.e., FLIP51, FLIP144, FLIP166, FLIP176, and FLIP196); G3—FLIP mutants showing a reduced colony diameter (i.e., FLIP16 and FLIP94); G4—FLIP mutants with a reduced growth phenotype that can be reverted under specific culture conditions (FLIP204 and FLIP149); and G5—FLIP mutants with a wet, purportedly autolytic, phenotype in the center of the colony under specific culture conditions (FLIP199). Group G6 corresponds to those FLIP mutants not included in any of the previous five categories (not shown). Scale bar = 2 cm.

**Figure 3 cells-08-01520-f003:**
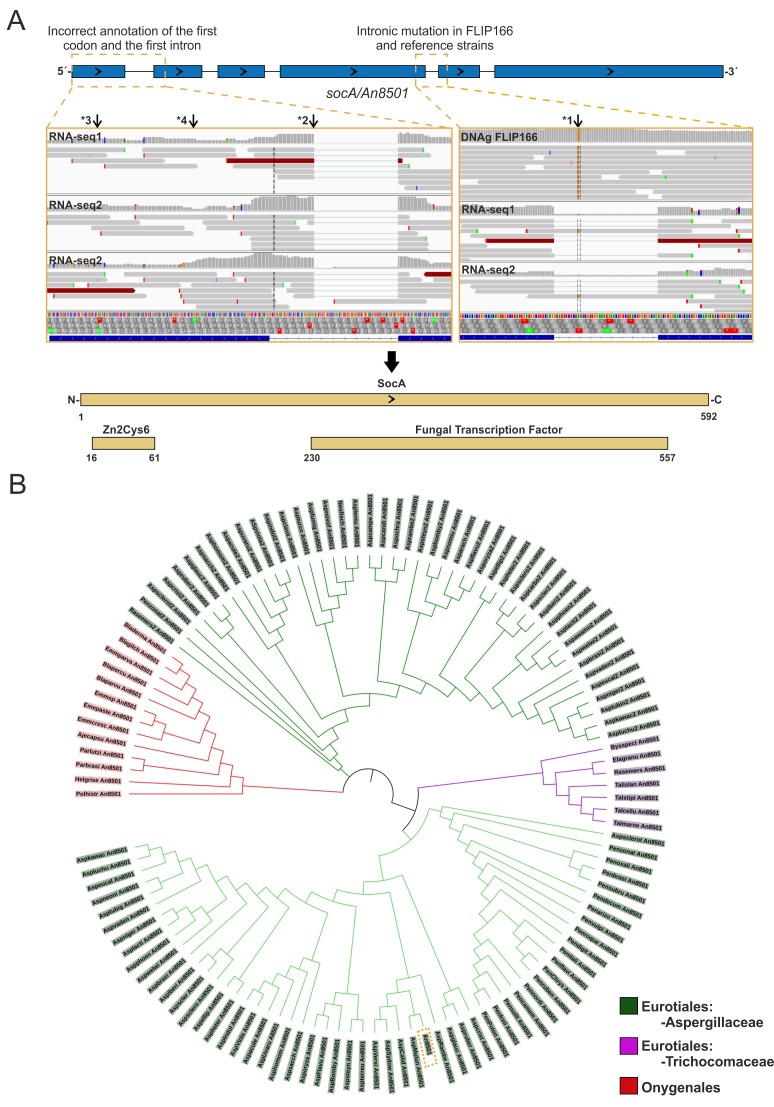
Analysis of *socA* sequence and protein evolution: (**A**) Sequence analysis of *An8501/socA*. The predictions of the intronic and exonic regions done by the AspGD Aspergillus genome (http://www.aspgd.org/) and FungiDB databases (https://fungidb.org/fungidb/) (top) are compared to DNA-seq (FLIP166) and RNA-seq (Δ*flbB* and a wild-type reference) experiments (middle). On the one hand, the right panel shows that the intronic mutation identified in FLIP166 in the fourth intron of *socA* was also present in the RNA-seq reads mapping to this intron in the reference strains (arrow *1), suggesting that it is not responsible for the FLIP166 phenotype. On the other hand, the left panel shows that the initiation codon (arrow *3) and the first intron were incorrectly annotated. Arrow *2 indicates the end of the first exon, and arrow *4 indicates the correct initiation codon. The bottom panel shows the prediction of the functional domains done by the InterPro website for the corrected amino acidic sequence of SocA, which includes two transcriptional regulatory domains. (**B**) Phylogenetic tree of SocA orthologs: The tree was built using MegaX (neighbor-joining method and 2500 replicates) and edited using iTOL. Red color designates SocA orthologs of species of the order Onygenales, while green and purple differentiate SocA orthologs of Aspergillaceae or Trichocomaceae species within Eurotiales, respectively. The dotted orange square marks the position of An8501/SocA in the tree. Note that the evolution of SocA includes a duplication event in specific *Aspergilli* and *Penicilli*.

**Figure 4 cells-08-01520-f004:**
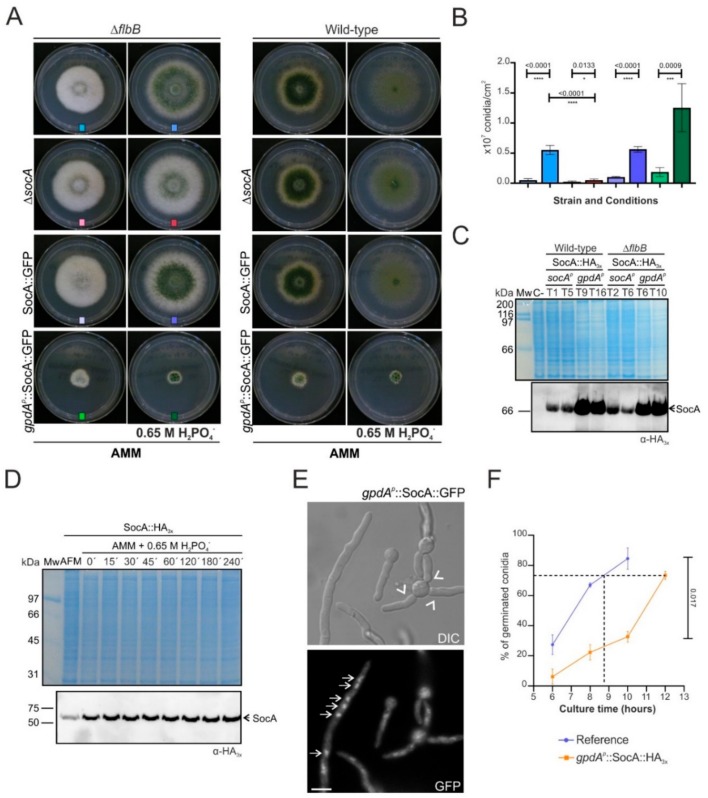
Functional characterization of SocA: (**A**) Phenotypes of Δ*socA*, SocA::GFP, and *gpdA^p^*::SocA::GFP strains, all in both Δ*flbB* (left) and wild-type (right) backgrounds, after 72 h of culture at 37 °C on AMM and AMM plus 0.65 M H_2_PO_4_^−^ (diameter of Petri dishes: 5.5 cm): The color code is the same as that used in Figure 4B. (**B**) Quantification of conidia production by the Δ*flbB* strains shown in panel A (bar colors keyed to photos): Values are given as the mean of at least three replicates plus SD. *p* values are given, and statistically significant (*p* < 0.05) relations are indicated with a variable number of asterisks (* *p* < 0.05; ***p* < 0.01; ****p* < 0.001; **** *p* < 0.0001). Double deletion of *flbB* and *socA* significantly decreases conidia production in AMM supplemented with H_2_PO_4_^−^ (**C**) Immunodetection of SocA::HA_3x_, driven either by native or *gpdA^p^* promoters, both in wild-type and Δ*flbB* backgrounds: Two clones per strains are shown. The Coomassie-stained gel is shown as a loading control. Driving expression through *gpdA^p^* clearly increases SocA levels. (**D**) Immunodetection of SocA::HA_3x_ (driven by the native promoter and in the wild-type *flbB* background) in mycelial samples collected after 15 h of culture in AFM and 0, 15, 30, 45, 60, 120, 180, and 240 min after the transference of mycelia to AMM supplemented with H_2_PO_4_^−^. The Coomassie-stained gel is shown as a loading control. Addition of 0.65 M H_2_PO_4_^−^ to the culture medium does not increase SocA levels. (**E**) Subcellular localization of *gpdA^p^*-driven SocA::GFP in germlings: The micrograph shows the altered germination pattern of the conidia (arrowheads), while fluorescence microscopy shows the nuclear localization of the putative transcriptional regulator (arrows). Scale bar = 5 µm. (**F**) Variation of the percentage of germinated conidia with the time of culture in liquid medium: Values of the *gpdA^p^*::SocA::HA_3x_ strain (orange line) are compared to a wild-type reference (purple line) and are given as the mean of two replicates plus SD. The dotted line shows that, in order to achieve the level of germination of the reference, the strain constitutively expressing SocA::HA_3x_ is delayed approximately three hours.

**Figure 5 cells-08-01520-f005:**
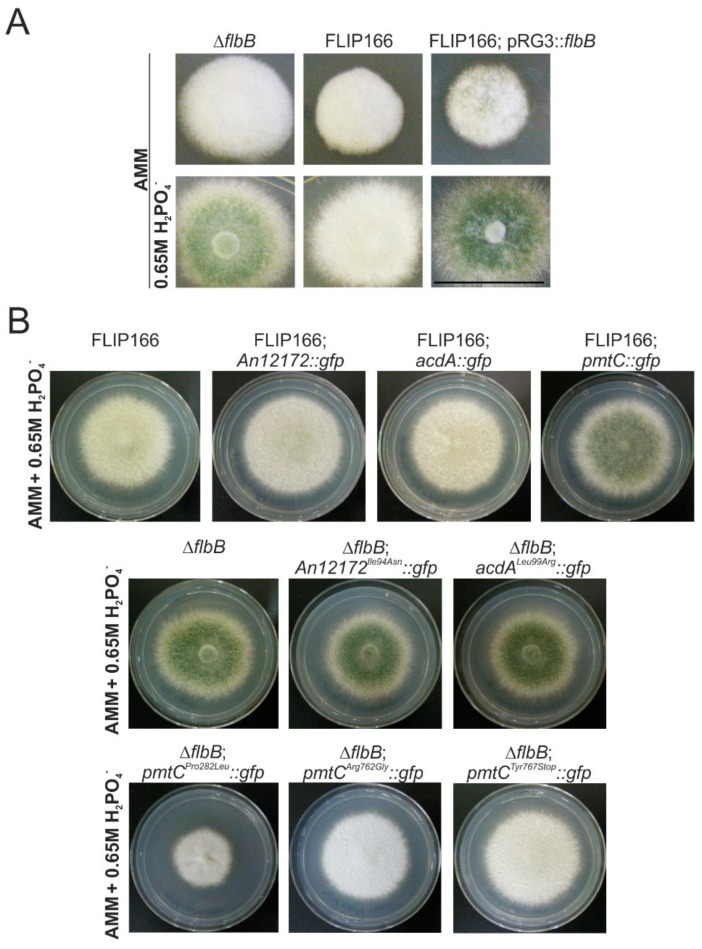
Identification of *pmtC* as the mutated gene responsible for the FLIP166 phenotype: (**A**) Phenotype of strains Δ*flbB*, FLIP166, and FLIP166 transformed with a p*RG3*::*flbB* plasmid after 48 h of culture on AMM (top row) and AMM supplemented with 0.65 M H_2_PO_4_^−^ (bottom row). Scale bar = 2 cm. The aconidial phenotype of FLIP166 under phosphate stress conditions is the result of the addition of the effect of *flbB* deletion and the unknown mutation. (**B**) Phenotypes of FLIP166 and FLIP166 transformed with *An12172*::*gfp*, *acdA*::*gfp* or *pmtC*::*gfp* constructs (row 1) or Δ*flbB* and Δ*flbB* transformed with the mutant versions of the abovementioned constructs (rows 2 and 3) after 72 h of culture on AMM supplemented with 0.65 M H_2_PO_4_^−^. Diameter of Petri dishes: 5.5 cm. Expression of a wild-type *pmtC* form suppresses the FLIP166 phenotype, while the insertion of a mutant *pmtC^Pro282Leu^* form in the null *flbB* background causes a FLIP phenotype with reduced growth. The two additional mutants of *pmtC* isolated (Δ*flbB* background), Arg762Gly and Tyr767Stop, show an aconidial phenotype on AMM supplemented with 0.65 M H_2_PO_4_^−^.

**Figure 6 cells-08-01520-f006:**
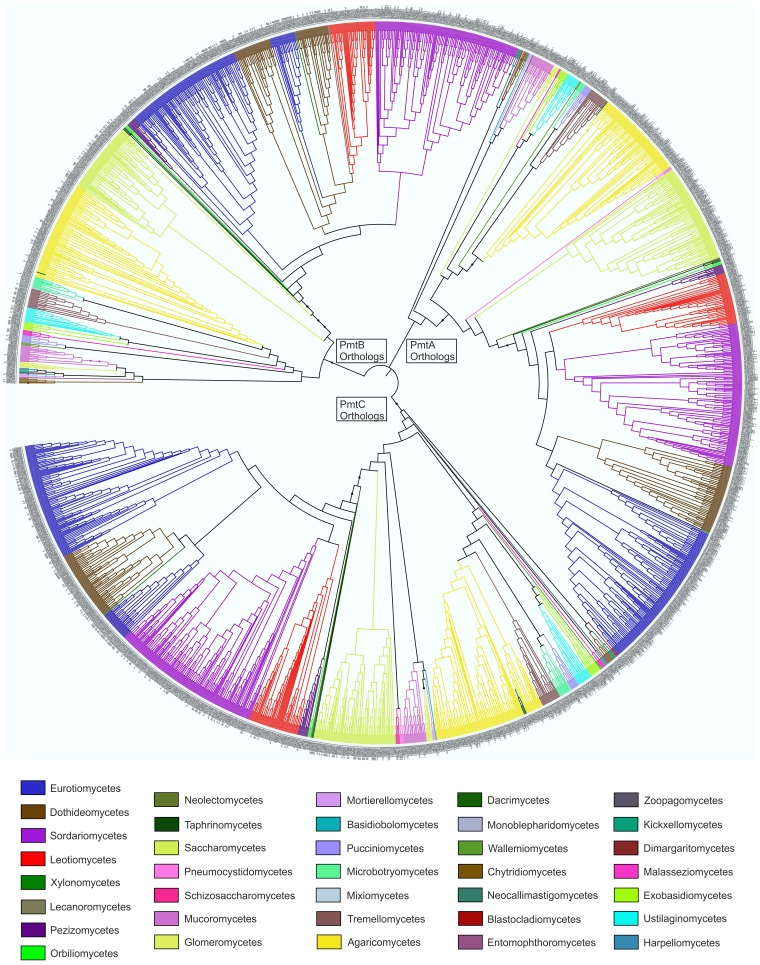
Evolutionary analysis of protein O-Mannosyltransferases of *A. nidulans*: Phylogenetic tree corresponding to 2169 fungal orthologs of *A. nidulans* PmtA/An5105, PmtB/AN4761, and PmtC/An1459. The tree was built using MegaX (neighbor-joining method and 1000 replicates) and edited using iTOL. The color key indicates the fungal class each ortholog belongs to. The clades corresponding to PmtA/B/C orthologs are indicated. Results suggest that the three protein O-mannosyltransferases are conserved in all fungal classes.

**Figure 7 cells-08-01520-f007:**
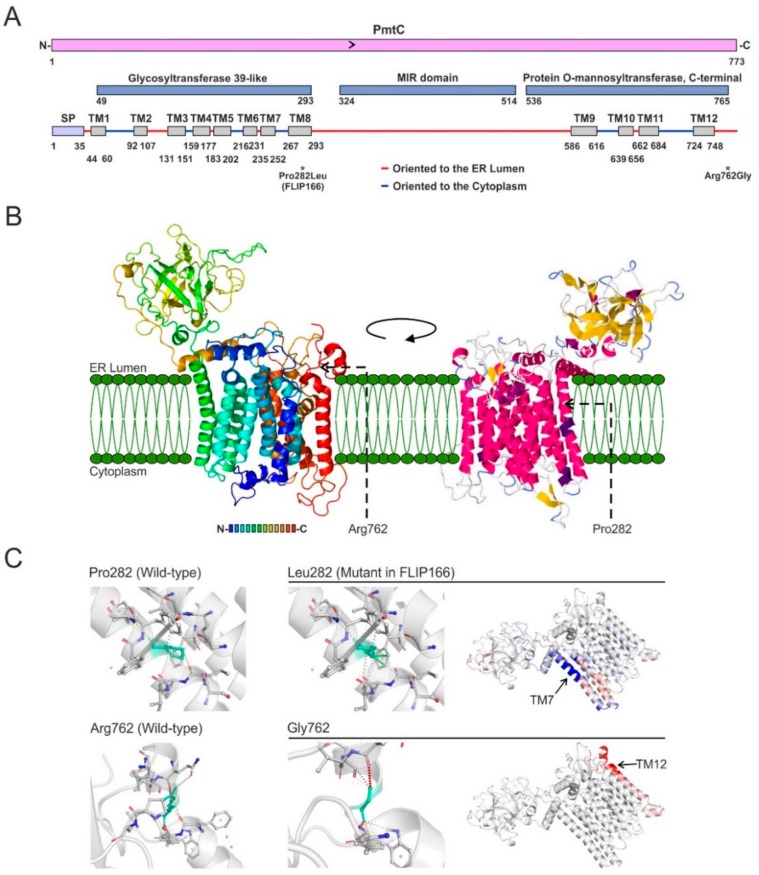
Domain organization and predicted structure of PmtC: (**A**) Domain analysis of PmtC based on InterPro and Phyre2. The extension of domains related to protein *O*-mannosyltransferase activity, as well as those of a signal peptide and the twelve transmembrane helices predicted are also indicated. Red and blue lines indicate orientation towards the ER lumen or the cytoplasm, respectively. (**B**) Prediction of the three-dimensional structure of PmtC carried out by Phyre2 and based on Protein Data Bank (PDB) entry 6P25. The rainbow-colored form on the left indicates the N- (blue) and C-terminal (red) ends of the protein, while the form on the right represents domains with secondary structure. The position of residues Pro282 (transmembrane, TM, helix 8) and Arg762 (C-terminus) are also indicated. (**C**) Assessment of the effect on interatomic interactions and protein stability of the Pro282 (left) to Leu (middle) substitution identified in the mutant FLIP166 (row 1) or the Arg762 (left) to Gly (middle) substitution identified in the transformation of the parental Δ*flbB* strain (row 2): Blue and red colors in the pictures on the right represent a gain of rigidity (Pro282Leu) or flexibility (Arg762Gly), respectively. The analysis was carried out using the DynaMut website.

**Figure 8 cells-08-01520-f008:**
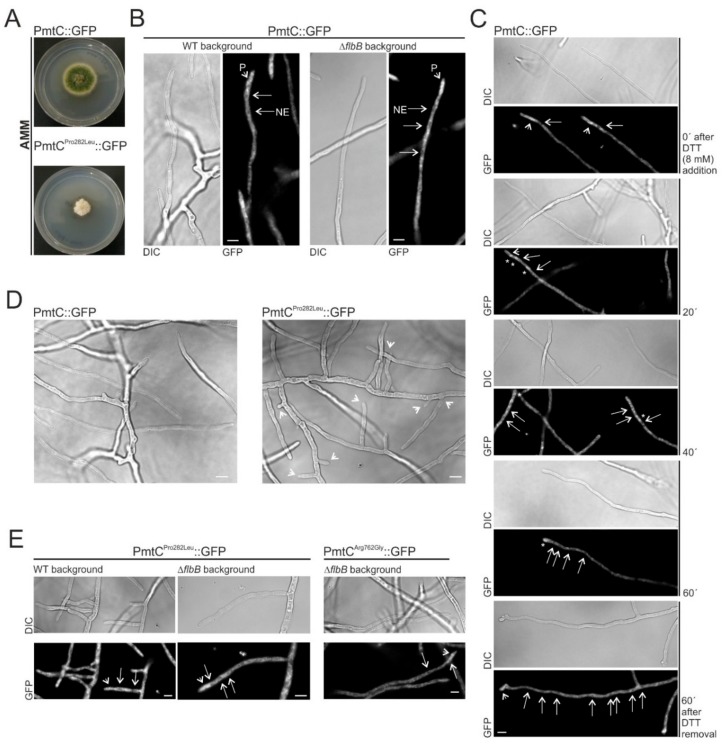
Subcellular localization of PmtC: (**A**) Phenotypes of strains expressing a wild-type or a mutant Pro282Leu form of PmtC::GFP (wild-type background) after 60 h of culture on AMM (diameter of Petri dishes: 5.5 cm). Substitution of Pro282 by a Leu causes an inhibition of radial growth and an aconidial phenotype. (**B**) Subcellular localization of PmtC::GFP in wild-type or Δ*flbB* hyphae: PmtC localizes to structures resembling the ER [52,53]. Scale bars = 5 µm. (**C**) Localization of PmtC::GFP in wild-type hyphae before and 20, 40, and 60 min after the addition of 8 mM DTT [52]. Recovery of growth and native PmtC localization after the removal of DTT is also shown (bottom panel). Scale bar = 5 µm. (**D**) Micrographs of hyphae of strains expressing PmtC::GFP or PmtC^Pro282Leu^::GFP chimeras (wild-type background). Arrowheads indicate the multiple adjacent branching sites observed in the mutant strain. (**E**) Subcellular localization of PmtC^Pro282Leu^::GFP (wild-type or Δ*flbB* backgrounds) and PmtC^Arg762Gly^::GFP (Δ*flbB* background) in hyphae. Both PmtC mutant forms maintain the localization of the wild-type form, suggesting that Pro282Leu or Arg762Gly substitutions affect PmtC activity but not its localization. Scale bar = 5 µm.

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
