# Peer review of "Identification and Characterization of *Aspergillus nidulans* Mutants Impaired in Asexual Development under Phosphate Stress"

_cells, 2019, doi:10.3390/cells8121520_

Round 1

Reviewer 1 Report

The manuscript by Otamendi et al. reports the functional characterization of Aspergillus Nidulans and developed the mutagenesis procedure.

This paper is interesting but lacking written and in some cases, the data presented poorly. However, I would recommend accepting this paper after a minor revision on the comments below.

(1) The abstract is not clearly written according to the paper’s topic and it is required to rewrite for a better understanding of the readers.

(2) As I guess, the authors used SDS-PAGE in Figure 4C and 4D but they didn’t describe why they used SDS-PAGE or whatever the aim to use it. If they used SDS-PAGE, reference is required. What are the bands visible on the gels? Please explain and let the readers know the relation of these protein bands with the findings of this paper. 

Author Response

The manuscript by Otamendi et al. reports the functional characterization of Aspergillus Nidulans and developed the mutagenesis procedure.

This paper is interesting but lacking written and in some cases, the data presented poorly. However, I would recommend accepting this paper after a minor revision on the comments below.

-The text has been carefully reviewed and we have considered the possibility of reorganizing the figures. We found it necessary in the case of Figure 7, which has been divided into new Figures 7 and 8. We believe that the work done has significantly improved the overall quality of the manuscript.

(1) The abstract is not clearly written according to the paper’s topic and it is required to rewrite for a better understanding of the readers.

-The abstract has been rewritten according to the recommendation of both reviewers. See lines 15-29 of the updated version of the manuscript.

(2) As I guess, the authors used SDS-PAGE in Figure 4C and 4D but they didn’t describe why they used SDS-PAGE or whatever the aim to use it. If they used SDS-PAGE, reference is required. What are the bands visible on the gels? Please explain and let the readers know the relation of these protein bands with the findings of this paper.

-In the legend of Figure 4, we stated for panels C and D that “The Coomassie-stained gel is shown as a loading control.” (lines 357 and 362). Nevertheless, and following the recommendation of reviewer 1, the main text has been modified to explain that the bands visible in the Coomassie-stained gels simply correspond to the proteins present in the protein extracts of Aspergillus nidulans and that these gels are shown as loading controls (lines 332-333 in the new version). A reference to SDS-PAGE (Laemmli, 1970; reference number 43) has been included in section 2.6, line 185. We didn´t include it in the first version because polyacrylamide gel electrophoresis is a standard technique in most molecular biology laboratories.

Submission Date

25 October 2019

Date of this review

05 Nov 2019 06:35:05

Reviewer 2 Report

This is an intersting paper shedding light on two new players taking part in the regulation of conidiation in A. nidulans, the putative transcription factor SocA and the O-mannosyltransferase PmtC, which is an "old chap" revisited by the Authors in this paper. 

The manuscript relies on a really huge amount of experimental work, which is very convincing for the Reader.

Nevertheless, the two separate parts of the paper with SocA multicopy suppressor of the FLIP166 phenotype in the first part and the functional analysis of the mutant form of PmtC responsible for the FLIP166 pheotype itself in the second have remained loosely coupled and this obviously made even the Discussion part less coherent. Not suprisingly, the Abstract itself reflects this kind of incoherence, which should be corrected by all means. In the Abstract, well-know facts are repeated about BrlA and FlbB and the new findings are compressed into one mere sentence. 

Table 1 should be transferred into the Supplement.

In section "Mutagenesis procedure" add light dosage, e.g. radiant intensity.

It is still not clear whether H2PO4- initiated stress will bypass the need for FlbB. Can you add more convincing evidence on this?

line 427: ...additional mutations in FLIP166 partially mask the real effect of ... the whole genome is in your hands, do you have at least a hypothesis on this?

Please add an overview (e.g. plot a Figure) on how these new players may interact with previously known UDAs. 

The English of the paper should be polished further. Some parts need a critical reading by a native speaker.

Minor remarks: line 15: ...thousands of asexual spores... There are much more.

line 39: ...kingdom fungi... write ....Kingdom Fungi....

line 39: ....miilions of species.... we have got more precise estimations in our hands

Author Response

This is an intersting paper shedding light on two new players taking part in the regulation of conidiation in A. nidulans, the putative transcription factor SocA and the O-mannosyltransferase PmtC, which is an "old chap" revisited by the Authors in this paper.

The manuscript relies on a really huge amount of experimental work, which is very convincing for the Reader.

Nevertheless, the two separate parts of the paper with SocA multicopy suppressor of the FLIP166 phenotype in the first part and the functional analysis of the mutant form of PmtC responsible for the FLIP166 pheotype itself in the second have remained loosely coupled and this obviously made even the Discussion part less coherent. Not suprisingly, the Abstract itself reflects this kind of incoherence, which should be corrected by all means. In the Abstract, well-know facts are repeated about BrlA and FlbB and the new findings are compressed into one mere sentence.

-The abstract has been rewritten following the recommendation of both reviewers (lines 15-29 of the new version of the manuscript), trying to balance the extension of the part introducing UDAs and CDPs and that corresponding to the new findings. The text has also been reviewed in a trial to improve the general coherence and strengthen the link among the different sections of the manuscript. However, in our opinion, there is no functional relationship between SocA and PmtC in the control of growth and conidiation. If both are characterized in this manuscript, it is because they have been identified in the different experimental approaches followed by us in the characterization of the mutant FLIP166. The manuscript has been structured to highlight the validity and usefulness of our strategy, the phenotypes analyzed and our experimental approaches for their characterization, not a hypothetic functional link between PmtC and SocA

Table 1 should be transferred into the Supplement.

-In agreement with reviewer 2 we have moved Table 1 to Supplementary material (now Table S1; oligonucleotides used are in Table S2).

In section "Mutagenesis procedure" add light dosage, e.g. radiant intensity.

-Included. Intensity of the lamp is 1,350 µW/cm2 (line 143).

It is still not clear whether H2PO4- initiated stress will bypass the need for FlbB. Can you add more convincing evidence on this?

-We meant that ∆flbB colonies conidiate profusely when they are cultured on a medium containing high H2PO4- concentrations, and that this suggests that the need for FlbB activity is bypassed under these conditions. The sentence has been modified in the abstract (lines 20-22) and the introduction (lines 70-72).  The phenotype of the null flbB strain on medium with 0.65 M H2PO4- is clear and reproducible, and is in our opinion enough to make this statement.

line 427: ...additional mutations in FLIP166 partially mask the real effect of ... the whole genome is in your hands, do you have at least a hypothesis on this?

-We didn´t include a discussion on that not to lengthen the manuscript unnecessarily with a speculative dissertation. Specific mutations identified in the mutant FLIP166 are located within exonic regions of genes coding for proteins predictably involved in the control of growth, such as for example AN0560 in chromosome VIII (see File S1). In this case, a lysine is replaced by a glutamic acid in position 488, just within the Exocyst component 84 C-terminal domain (PF16528). However, this mutation would suppress partially the growth phenotype caused by the PmtCPro282Leu mutation in FLIP166 and would be thus a gain-of-function mutation. That should be confirmed experimentally in the future and that is why we believe that introducing this piece of text in the updated version of the manuscript would be too speculative.

Please add an overview (e.g. plot a Figure) on how these new players may interact with previously known UDAs.

-In our opinion, the possibility of including a figure with a model functionally linking the activity of UDAs or CDPs with SocA or PmtC should be considered with caution. As discussed in the manuscript, their role in conidiation is probably indirect (SocA; lines 568-571 and 584-586) or may be a consequence of a central role in growth (PmtC), since polar growth is required for example in specific stages of metulae and phialide formation (lines 617-627). Consequently, we believe that a model such as that proposed by the reviewer would be biased and would lead to misinterpretation.

The English of the paper should be polished further. Some parts need a critical reading by a native speaker.

-The text has been reviewed and modified (see the updated file).

Minor remarks: line 15: ...thousands of asexual spores... There are much more.

-Yes, there are more. However, we have deleted this sentence in the new version of the abstract.

line 39: ...kingdom fungi... write ....Kingdom Fungi....

-Modified (line 40).

line 39: ....miilions of species.... we have got more precise estimations in our hands

-Modified with the estimate of Hawksworth and Lücking in Microbiology Spectrum (2017) (line 40).

Submission Date

25 October 2019

Date of this review

18 Nov 2019 17:49:13